# A Core Transcription Regulatory Circuitry Defining Microglia Cell Identity Inferred from the Reanalysis of Multiple Human Microglia Differentiation Protocols

**DOI:** 10.3390/brainsci11101338

**Published:** 2021-10-11

**Authors:** Antoine Aubert, François Stüder, Bruno Maria Colombo, Marco Antonio Mendoza-Parra

**Affiliations:** Génomique Métabolique, Genoscope, Institut François Jacob, CEA, CNRS, Univ Evry, Université Paris-Saclay, 91057 Evry, France; Antoine.AUBERT@genoscope.cns.fr (A.A.); fstuder@genoscope.cns.fr (F.S.); bruno.colombo@univ-evry.fr (B.M.C.)

**Keywords:** microglia differentiation, transcriptomics, gene regulatory networks, systems biology

## Abstract

Microglia, the immune cells in the brain involved in both homeostasis and injury/infection control, play a predominant role in neurodegenerative diseases. In vivo studies on microglia are limited due to the requirement of surgical intervention, which can lead to the destruction of the tissues. Over the last few years, multiple protocols—presenting a variety of strategies—have described microglia differentiation issued from human pluripotent stem cells. Herein, we have reanalyzed the transcriptomes released on six different microglia differentiation protocols and revealed a consensus core of master transcription regulatory circuitry defining microglia identity. Furthermore, we have discussed the major divergencies among the studied protocols and have provided suggestions to further enhance microglia differentiation assays.

## 1. Introduction

Microglia represent about 10% of the cells in the brain. They are localized in the parenchyma and are the most representative immune cells of the central nervous system (CNS) [1]. Microglia derive during early embryogenesis from yolk sac–primitive macrophages and begin to colonize the brain before the emergence of neurons and macroglia. These primitive myeloid progenitors are highly proliferated throughout embryonic life and constitute the resident adult microglia in a healthy brain. These cells persist during adulthood via constant self-renewal. Nevertheless, postnatal hematopoietic progenitors also contribute to the adult microglial pool and are constantly renewed throughout life [2,3,4].

Microglia are highly dynamic cells that constantly monitor the brain microenvironment. These cells guarantee cell homeostasis by ensuring tissue repair. Once stimulated, microglial cells rapidly proliferate, change morphology, and activate and secrete various immunomodulatory factors (proinflammatory cytokines) [5,6,7].

Thanks to their ability to scan the microenvironment in the brain, microglia are also involved in the recognition and the elimination of pathogens invading the CNS [8,9,10] and are expected to play a predominant role in the context of neurodegenerative diseases, including Alzheimer’s disease (AD) [11,12,13,14,15], Parkinson’s disease (PD) [16], and progressive supranuclear palsy (PSP), which corresponds to a common type of atypical parkinsonism that is based on a four-repeat tauopathy pathology [17,18]. Concerning AD, there is growing evidence that microglia protect against the incidence of this illness, as suggested by the observation of their accumulation around β-amyloid plaques. Furthermore, recent genome-wide association studies (GWAS) have revealed an astonishing accumulation of AD risk loci near or within genes known to be expressed or that are even sometimes specifically in microglia, which suggests a genetic dimension that directly influences microglia behavior in the context of AD [19]. Similarly, recent studies have suggested that human endogenous retroviruses (HERV) might contribute to microglia activation, in both neurodegenerative diseases as well as in neuropsychological disorders, by regulating the expression of inducible NOS [17,20,21].

Motivated by the necessity of studying human microglia behavior in a normal setting as well as in the context of neurodegenerative diseases, over the last six years, multiple protocols described microglia differentiation strategies issued from human pluripotent stem cells [22]. In most of these studies, the obtained iPSC-derived microglia-like cells (iMGL) were validated by phenotypical assays as well as by the use of comprehensive transcriptome readouts (microarrays [23,24] or RNA-sequencing [25,26,27,28,29,30]), allowing for the investigation of the presence of the signature genes of microglia. While the authors of these studies systematically found relevant microglia-related expressed genes, a systematic comparison between all of the transcriptomes assessed within microglia differentiation studies is missing.

In this article, we provide a detailed comparative study—covering six different protocols—evaluating (i) the quality of the generated transcriptome data; (ii) their relevant gene expression differences; (iii) the presence of a consensus gene regulatory program among the compared studies as well as their particularities in the context of master transcription factors.

Overall, this comparative study reveals a core gene regulatory program defining the microglia state and provides evidence for the differences observed between the studied protocols. Notably, it provides new avenues to explore master transcription factor over-expression-driven strategies for enhancing microglia differentiation methods, which are supported by the fact that master transcription factors possess the capacity to trigger a developmental program outside of its normal context [31].

## 2. Materials & Methods

### 2.1. Public Transcriptomes

Datasets associated with the six protocols under study were collected from the Gene expression Omnibus (GEO) database. The corresponding articles and their associated GEO series accession number (GSE) are Abud et al. (2017) [26] (GSE97744); Muffat et al. (2016) [32] (GSE85839); Douvaras et al. (2017) [27] (GSE97744); McQuade et al. (2018) [26] (GSE117829); Trudler et al. (2021) [29] (GSE169065); and Chen et al. (2021) [30] (GSE163984).

### 2.2. Data Pre-Processing

Raw fastq files were aligned to the reference human genome GRCh38 using Bowtie2 (v2.4.1). Aligned datasets were qualified for read count saturation using the next generation sequencing quality control generator (NGS-QC Generator) [33,34]. Furthermore, read counts associated with annotated transcribing regions were assessed with the tool featureCounts (v2.0.1) [35] and were assembled within a matrix composed by genes in rows and all datasets (issued from all protocols) in columns. The raw matrix was cleaned by removing genes devoid of read counts, and then one read was added over the whole matrix to avoid division by zero errors.

Finally, technical amplitude differences between compared datasets were corrected by applying a quantile normalization procedure. Data sets were regrouped by cell type and genetic background into subgroup or replicate datasets in each protocol. Each replicate dataset was associated with a number. All of the datasets used in this study are summarized in Appendix A.

### 2.3. Principal Component Analysis

A principal component analysis (PCA) was performed on the log10 value quantile-normalized counts. Specifically, PCA was performed on genes that appeared as upregulated on in vivo fetal microglia provided by Abud et al. (6319 genes). The three-dimensional PCA analysis was plotted using the “plot3D” function of the rgl package.

### 2.4. Differential Gene Expression Analysis

Differential gene expression analysis between samples was performed using Deseq2 (v.1.32.0) on the quantile normalized count matrix. Datasets associated with a given protocol were compared to their corresponding iPSC readouts when available. When the iPSC data were not provided (Muffat et al. (2016), Douvaras et al. (2017), and McQuade et al. (2018) data sets), the iPSC datasets provided by Abud et al. (2017) were used as a reference for the differential expression analysis. Differentially expressed genes were considered for fold-changes > 2 and presenting an adjusted *p*-value (padj) < 0.01 (Appendix A)

To determine, the composition of the microglia identity of each iMGL population, upregulated genes were compared to their in vivo counterpart (fetal and adult) to determine the percentage of commonly up regulated genes between each iMGL population compared to two in vivo controls (fetal and adult microglia from Abud et al.). This value of similarity was tempered by the divergence between the in vivo microglia and the iMGL (the percentage of up regulated genes in iMGL that were not differently expressed in their in vivo counterpart). This value allowed us to take the variability of the number of up regulated genes between samples into account.
$$\%Commun\;genes=\frac{Number\;of\;commun\;upregulated\;genes\;}{Number\;of\;upregulated\;genes\;\left(MGL\right)}\times 100\;\%Divergence=\left(1-\frac{Number\;of\;commun\;upregulated\;genes\;}{Number\;of\;upregulated\;genes\;\left(iMGL\right)}\right)\times 100$$

Microglia-specific markers were defined by selecting upregulated genes that were present in the fetal microglia dataset (6154 genes) but that were absent from all other control samples (CD14+/CD16- monocytes, CD14+/CD16+ inflammatory monocytes, myeloid dendritic cells, neuronal precursors, and neuronal-related datasets). This Microglia-specific marker subset (201 genes; Appendix A) was further confirmed by gene ontology analysis using ENRICHR [36] and was used to profile gene expression changes within the iMGL samples. In each case, datasets were clustered by performing Euclidian biclustering. Heatmaps were plotted using the “pheatmap” package on R.

### 2.5. Reconstruction of Gene Co-Regulatory Networks

Differential gene expression readouts (differential log2 fold-change levels higher than 2; smaller than −2) associated with each of the studied protocols were integrated with transcription factor-target gene relationships inferred from human CAGE readouts assessed by the FANTOM consortium covering more than 300 human cell/tissue types [37]. This integrative processing was performed within TETRAMER, which also allowed us to infer master regulator TFs by modelling transcription signaling regulation propagation [38]. TETRAMER allowed us to rank TFs on the basis of their capacity to regulate down-stream target genes that are associated with each of the studied protocols. Ranked TF-TG co-regulatory networks issued from all six protocols were integrated into a single network. Gene ontology analysis on the common master regulatory TFs was performed using ENRICHR [36].

Inferred consensus master transcription factors were compared with master players described in three further studies: Gosselin et al. [39], which was based on in vivo microglia isolated from surgically resected brain tissue and that had been cultured in an in vitro environment; Banerjee et al. [40], who described a protocol for iMGL generation from peripheral monocyte samples; and Grubman et al. [41], who described an enhanced maturation of iMGL precursors when co-cultured with human neuronal precursors (ReN) (Appendix A). Specifically, the raw read counts provided by Grubman et al. (GSE125872) were used to query for genes being over-expressed in the iMG + ReN conditions relative to their corresponding iPSC sample. Similarly, the microglial signature genes provided by Banerjee et al. in their Appendix A were used for this comparison. Finally, master transcription factors associated with the study of Gosselin et al. were collected from Appendix A retrieved in Banerjee et al. [40] as well as from the information provided on the corresponding publication.

## 3. Results

### 3.1. Comparative Reanalysis of Whole Transcriptome Readouts Generated within Six iPSC–Derived Microglia (iMGL) Differentiation Studies

Concerned by the variety of strategies proposed for generating microglia cells from human iPSC differentiation and their potential similarities and discrepancies on their performance, we have focused our attention on six protocols, namely the 70-day long protocol provided by Muffat and co-authors [32]; the protocol proposed by Abud and co-authors that based on a hematopoietic intermediate but that also required hypoxic conditions [25] as well as its improved version relying on a commercial solution issued from a stem cell technologies company [26]; the protocol proposed by Douvaras and colleagues that is based on an initial myeloid differentiation [27]; the 21-day protocol proposed by Trudler et al. that mimics normal yolk sac and erythromyeloid embryonic development [29]; and the extremely fast protocol (10 days) proposed by Chen et al., which is focused on the forced expression of the master transcription factors SPI1 (SFFV pro-viral integration 1) and CEBPA (CCAAT Enhancer Binding Protein Alpha) [30] (Figure 1A).

Each of these studies generated whole transcriptome readouts associated with the iMGL state, their hematopoietic intermediates (HPC) when available, their related iPS precursors as well as positive controls issued from fetal and adult microglia cells, and negative control samples corresponding to CD14+/CD16− monocytes (CD14M), CD14+/CD16+ inflammatory monocytes (CD16M), myeloid dendritic cells (Blood DCs), neuronal precursors (NPC), and neuronal transcriptome data. Overall, as part of this comparative analysis, we collected 148 transcriptomes collected from the public gene expression omnibus (GEO) repository and reprocessed them with the help of our in-house qcGenomics pipeline [33].

As part of the primary analysis, we evaluated the quality of the collected datasets. This was performed using our previously described QC STAMP-scoring methodology, which assess the sequencing-depth saturation point by comparing the level of read-count pattern reproducibility between the original RNA-seq profile and those where only a subset of the total mapped reads are used for reconstruction [34,42]. As illustrated on Figure 1B, most of the reanalyzed datasets present QC STAMP scores comprised between 15 and 25 despite of their mapped reads, while the transcriptomes generated by Douvaras and colleagues present a relatively lower quality scoring level. In all cases, the assessed QC STAMP scores are within the highest levels (quality A) estimated over more than 90,000 public datasets (see our database http://ngs-qc.org/database.php (accessed on 1 October 2021) and Appendix A), indicating that the processed RNA-seq readouts are of high quality.

### 3.2. iMGL Transcriptome Datasets Issued from Different Protocols Present Variable Degree of Similarity with Control Microglia Samples

With the aim of evaluating the microglia differentiation performance between the compared protocols, we have first evaluated the gene expression levels of well-known microglia markers within the analyzed transcriptomes. As illustrated on Figure 2, significant gene expression is observed for the G protein-coupled receptor GPR34 and the transcription factor SPI1 in all iMGL transcriptomes, which is in agreement with the transcription levels observed on the fetal and adult microglia readouts. On the contrary, expression differences are observed for the gene CD45 and TREM2, where no signals were observed in datasets generated by Chen et al. Similarly, TMEM119 is devoid of readouts on the McQuade and Abud’s iMGL datasets; P2RY12 do not present read counts on Muffat, Douvaras, Chen and Trudler’s iMGL datasets, and the expression of the CX3CR1 gene can be seen in the Abud and McQuade iMGL data.

These major differences observed on a small subset of genes strongly support the need for large scale analyses able to decrease potential technical differences and to highlight the relevance of each of these protocols for generating iMGL cells. For this, a principal component analysis (PCA) was used to compare all of the iMGL datasets as well as the various positive and negative control samples (Figure 3A). The use of a 3-dimensional PCA representation (PC1: 60.9%; PC2: 10.4%; PC3: 6% variance, respectively) allowed us to reveal the similarities between the microglia control datasets and the iMGL samples issued from Muffat, Douvaras, and Troudler’s protocols, which are located far away from the human iPS, neuronal precursors (NPCs), and neurons as well as the hematopoietic datasets (CD14, CD16, Blood DC). Surprisingly the iMGLs issued from Chen’s protocol appeared distant from the microglia control datasets and rather proximal to the hematopoietic precursor (HPC) and hematopoietic datasets. Similarly, most of the iMGLs issued from Abud and McQuade’s protocols appeared to be preferentially distal from microglia control samples, proximal to HPCs, but distant from hematopoietic samples, suggesting that they are more similar to microglia than hematopoietic samples.

To further evaluate the potential differences between the transcriptomes associated with the various iMGL samples, a differential expression analysis was performed relative to their corresponding induced pluripotent stem cell precursors (Appendix A). The analysis of the number of upregulated genes (2-fold change; adjusted *p*-value < 0.01) demonstrated > 55% of commonalities between most of the iMGL replicates issued from McQuade, Muffat, Douvaras, and Trudler’s protocols and the microglia control samples (Figure 3B). Indeed, only one of the iMGL replicates issued from the Douvaras’ protocol shared less than 50% of their upregulated genes with the microglial control samples (fetal and adult). Surprisingly, the iMGL samples issued from Chen’s protocol only shared ~30% of their upregulated genes with the microglia control samples, which is in agreement with the low similarity observed in the PCA analysis. Furthermore, the fraction of upregulated genes that diverge relative to either the fetal or adult microglia samples revealed that iMGL samples issued from Chen’s protocol presented the highest discrepancies (~58%).

To take in consideration the fold change levels when performing these comparative analyses, we first selected the upregulated genes within the microglia control samples and then subtracted the upregulated gene commonly expressed in negative control samples (CD14M, CD16M, Blood DC, NPC and Neurons) (Appendix A). This set of 201 genes also appeared to be strongly upregulated (>4-fold change) in the iMGL samples issued from Muffat, Douvaras, and Trudler’s protocols, where ~70% to 80% of the microglial genes were upregulated (Figure 3C and Appendix A). The iMGL samples issued from Abud and McQuade’s protocols presented 36% upregulated genes, while the iMGL samples issued from Chen’s protocol only presented 23% of these selected microglia-specific upregulated genes. Remarkably, one of the iMGL replicate datasets issued from the protocol described by McQuade et al. (McQuade.5) showed a strong similarity with the fetal and adult microglia control samples. On the contrary, one of the replicates generated by Douvaras et al. (Douvaras.1) showed a lowered similarity to in vivo control samples (Figure 3C). A gene ontology analysis performed on the set of 201 genes confirmed their enrichment on microglia-related terms at a high confidence level (Figure 3D).

### 3.3. A Master TF Co-Regulatory Network Driving Microglia Differentiation Reveal Strengths and Weaknesses of the Compared Protocols

Microglia differentiation, when induced by whatever protocol, might reshape the gene regulatory programs defining stem cell towards a microglial state. To reconstitute the gene regulatory programs involved in microglia differentiation, we have integrated the differential iMGL gene expression readouts with TF-TG relationships issued from the integration of human CAGE readouts assessed by the FANTOM consortium covering more than 300 human cell/tissue types [37]. This integrative effort has been performed with TETRAMER, our previously developed Cytoscape application [38], leading to the reconstruction of iMGL-related gene regulatory networks associated with each of the compared protocols. Considering that the reconstructed networks present TF–TG relationships issued from a variety of cell/tissue types, TETRAMER applies an in silico transcription regulation signaling cascade initiated from each of the TF to conserve those coherent with the assessed gene expression information related to the microglial state. This in silico modeling strategy allows us to compute a master regulatory index for each of the TFs that is defined as the fraction of target genes (direct and indirect) that can be controlled by a given TF relative to the total number of nodes within the GRN (Appendix A).

Assuming that all six protocols are expected to drive towards the same microglial state, we compiled all of the TFs presenting significant MRI scores within all of the protocols and selected those that are present on at least three of the six cases. This analysis led to 57 TFs, which were classified into three groups. The first group is composed of 19 TFs that were present on all six protocols. The second subset is composed of 22 TFs expressed on the transcriptomes issued from most of the protocols, except the one described by Chen and colleagues. Finally, the third group is composed of 16 TFs that were upregulated on transcriptomes issued from the protocol provided by Chen and colleagues as well as some of the other protocols (Figure 4A). A gene ontology enrichment analysis performed for each of these groups revealed that only the master TFs retrieved on all of the protocols (first group) or those retrieved on protocols other than that using specific TFs-overexpression (second group) are significantly enriched for microglia-related terms (Figure 4B). This microglia-related TFs enrichment is also supported by a comparison with master players retrieved in three other studies, either issued from in vivo microglia collected from surgically resected brain tissue [39], from matured microglia due to their co-culturing with human neuronal precursors [41], or differentiated from peripheral monocytes [40] (Appendix A).

Considering that the specific TF-overexpression protocol used by Chen and colleagues targets the factors CEBPA and SPI1, we focused our attention to their corresponding regulome. As illustrated in Figure 4A, both CEBPA and SPI1 are retrieved over-expressed in all of the protocols and present strong master regulatory indexes (SPI1: MRI > 70%; CEBPA: MRI > 30%). Their related TF co-regulatory network reveals an important number of co-regulated master players, including MAF, FLI1, IRF2, RUNX3, and the cellular receptor for Vitamin D3, VDR; all of them retrieved among all six protocols (Figure 4C).

Interestingly, another group of SPI1/CEBPA co-regulated players, including IRF9, MEF2A, NFIC, IRF8, CEBPB, EGR2, REL, and ETV6, were found expressed in most protocols, with the exception of the one described by Chen and colleagues (Figure 4A,C). Specifically, the transcription factor IRF8 (interferon regulatory factor 8), known to be essential for proper microglia reactivity [43] as well as for its development [44], and the NF-κB family member REL (c-rel), known to be expressed in the brain and playing relevant roles in protection from neurodegenerative-associated stimuli [45], appeared to be co-regulated by SPI1 and CEBPA but not expressed within the protocol described by Chen et al. (Figure 4C). Other master TFs such as CEBPB, which was previously described as a major transcription factor regulating the expression of genes involved in inflammatory responses [46], and the ETV6 transcriptional repressor, which is known to play a critical role in hematopoiesis [47], are described as being regulated by CEBPA within the reconstructed regulome but not expressed within the protocol described by Chen et al. (Figure 4C).

These cumulative observations strongly support for a suboptimal activation of the core transcription regulatory circuitry driving microglial differentiation when using the protocol described by Chen and colleagues, which is most likely due to the partial activity associated with the over-expressed factor CEBPA. In fact, the read count enrichment at the CEBPA locus observed in the RNA-seq samples issued from the CEBPA over-expression generated by the protocol described by Chen and colleagues, reveals a shorter gene expression version than that observed in RNA-seq samples issued from all other protocols as well as those related to the fetal and adult microglia control datasets (Figure 4D). This shorter CEBPA transcript coincides also with a lower molecular weight being reported by the authors (268 amino acids) relative to the molecular weight described in public repositories (358 amino acids).

## 4. Discussion and Conclusions

There is increasing evidence that microglia, the most performant immune cells in the brain, is involved in both homeostasis and injury/infection control and that it also plays a predominant role in neurodegenerative diseases. Indeed, microglia activation is dependent on neuroinflammation and might also be related to the HERV, which increases NO production [17]. This being said, studies on microglia in vivo are limited due to the necessity of surgical intervention, which can lead to the tissue destruction and the consequent activation of the microglia. Furthermore, their transfer to a tissue culture environment results in the rapid and extensive downregulation of microglia-specific genes [39]. Histological studies, on the other hand, may be helpful to provide clues regarding cell involvement in their environment; nevertheless, they are limited in terms of evaluating the functions of living cells [48].

In order to overcome this problem, different laboratories have moved towards the generation of microglia in vitro from circulating progenitors and stem cells. The first attempts successfully obtained microglia-like (MGL) cells from circulating monocytes and bone marrow-derived stem cells [49,50]. Nevertheless, these approaches represent non-ideal models since the microglia that were obtained do not share the same embryonic origin as the microglia in the brain, and these cells present a relatively advanced states of differentiation, making them unsuitable for asking questions about the regulatory networks governing their development. More recent approaches have focused on the differentiation of stem cells, either those issued from embryonic sources or from the reprogramming of differentiated material (ESC and iPSC, respectively) [22].

The first article, published in 2016, described a 74-days long protocol describing the differentiation of iPSC into microglia-like (iMGL) cells through the intermediate passage, generating neural progenitors [32]. Since then, several other protocols have been described [23,24,25,26,27,28,29], most of which are 30 days shorter than the previous one, notably because they do not require the neural progenitor step and because the generation of hematopoietic progenitor cells (HPC) can be achieved by mimicking the in vivo microglial cells ontogeny [3]. Finally, Chen and colleagues, described a novel approach that produces the differentiation of iMGL by overexpressing the transcription factors SPI1 and CEBPA, both of which are known to play a major role in microglial cell fate during embryogenesis [30].

The transcriptomic reanalysis effort described in this article allowed us to highlight the commonalities but also the differences in gene programs between all six of the compared protocols. Importantly, this effort revealed a core transcription regulatory circuitry driving microglial differentiation composed by 19 master TFs that were over-expressed on all protocols, including SPI1 and CEBPA (Figure 4A). A second subset of master regulators that were expressed on all protocols (IRF9, MEF2A, NFIC, IRF8, CEBPB, EGR2) except the one described by Chen and colleagues was observed. Importantly, these factors are also part of the reconstructed SPI1/CEBPA regulome, supporting their direct connection to the inner core transcription regulatory circuitry (Figure 4A,C).

While all of these findings fully support the hypothesis of Chen and colleagues, which indicates that microglia differentiation can be driven by over-expressing SPI1 and CEBPA, the major discrepancies between this last protocol and all others seems to have origin on a truncated version of the CEBPA factor used during such study. In this context, it would be essential to reconsider the strategy, this time using endogenous promoter activation systems, such as the use of RNA-guided gene activation by CRISPR-Cas9-based transcription factors [51].

Finally, the presented core transcription regulatory circuitry provides means to enhance microglia differentiation protocols, but also to evaluate potential deregulated core transcription regulatory circuits, which might arise in the context of neurodegenerative diseases, which, for instance, can be modeled with the help of 3-dimensional cerebral organoid systems.

## Figures and Tables

**Figure 1 brainsci-11-01338-f001:**
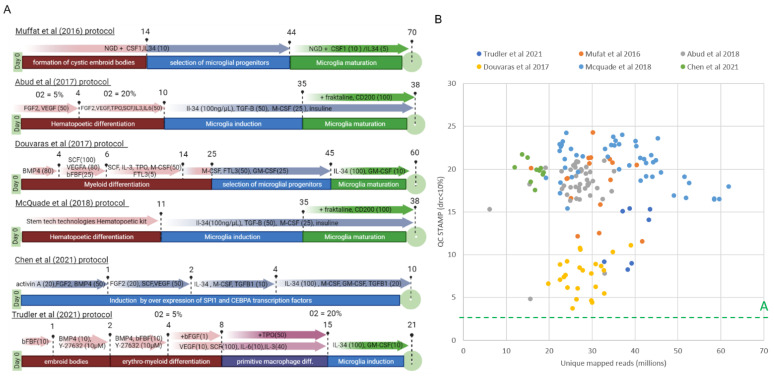
Overview of the iMGL protocols and the quality of their RNA-Seq libraries. (**A**) Short summary of the human iMGL differentiation protocols compared in this study. Protocols are organized by chronological order of publication in the literature. The different steps of the cell differentiation were highlighted and associated with the corresponding treatment (when not specified the unit’s concentration is ng/µL). Each time point corresponds either to a key step in the cellular differentiation or a change in medium composition. The O_2_% in the medium was also added when modified during the protocol. When not specified, the O_2_ concentration is considered to be normoxya (O_2_ = 20%). (**B**) Scatter plot of the quality of each RNA-Seq libraries associated with the compared protocols as assessed by the NGS-QC Generator tool compared to their corresponding unique mapped sequenced reads. The green dashed line demarcates the quality “A” threshold assessed by the NGS-QC Generator database, issued from the analysis of more than 90 thousand public datasets (http://ngs-qc.org/accessed on 1 October 2021).

**Figure 2 brainsci-11-01338-f002:**
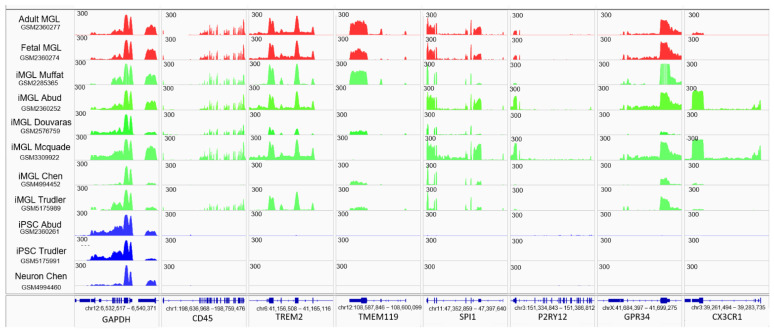
RNA-sequencing read coverage on microglial specific gene loci. Datasets corresponding to control in vivo microglia samples (adult and fetal cells) are shown in red, induced microglia-like (iMGL) datasets associated with each of the studied protocols are shown in green, and a composed negative control corresponding to the IPSC and neurons is shown in blue. To enhance their comparability, the read count amplitude for all samples has been set to 300.

**Figure 3 brainsci-11-01338-f003:**
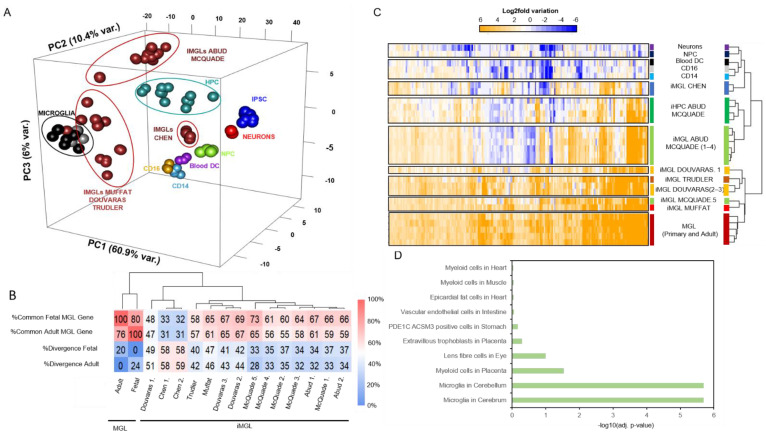
Differences and similarities between the iMGL gene expression profiles issued from six differentiation protocols. (**A**) Three-dimensional principal component analysis (3D PCA) showing iMGL in dark red, in vivo microglia (fetal and adult microglial cells in black and grey, respectively), induced hematopoietic progenitor cells (iHPC) in azure, CD14+/CD16− quiescent monocyte in green, CD14+/CD16+ inflammatory monocytes in yellow, blood dendritic cells in violet, neurons in red, and IPSC in blue. The first principal component (PC1 60.9% variance) represents the hematopoietic differentiation process, principal component 2 (PC2 10.4% variance) allows the separation of the in vivo microglia cells and the iMGL datasets, and principal component 3 (PC2 6% variance) allows the discrimination between the different monocytic population trajectories. The PCA graph reveals the stratification of all iMGL samples on three major clusters. The first cluster is composed by iMGL samples issued from protocols developed by Muffat et al., Douvaras et al., and Trudler et al. and presents a striking similarity with the in vivo microglia control samples. The second cluster, separated from the first cluster because of PC2, is composed by iMGL samples issued from protocols developed by Abud et al. and McQuade et al. Finally, the third cluster present in the monocytic region of the graph presented the iMGL samples generated by Chen et al. (**B**) Similarity and divergency matrix (in percent) for each of the iMGL datasets compared to the control microglia samples. This analysis is based on the upregulated genes (2-fold change; adjusted *p*-value < 0.01) assessed in each of the iMGL samples relative to the corresponding induced pluripotent stem cell (iPSC) precursor samples. (**C**) Gene expression levels (log2 fold-change, Euclidian clustered heatmap) assessed on iMGL and the corresponding control samples on a subset of 201 microglia-specific genes. (**D**) Cell ontology of the 201 microglia-specific genes.

**Figure 4 brainsci-11-01338-f004:**
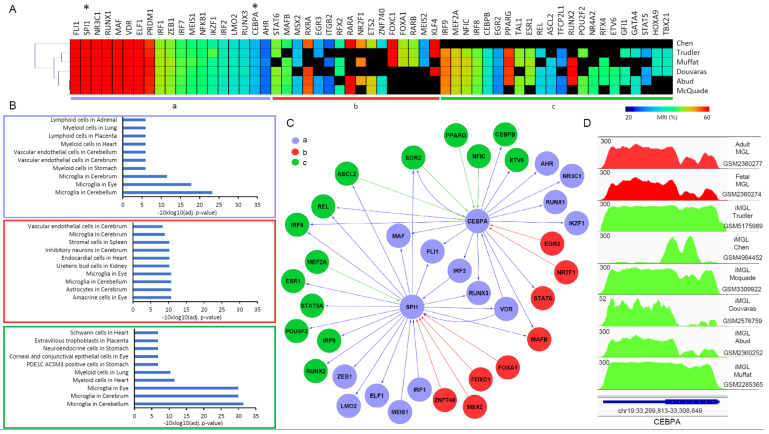
In silico prediction of a consensus master transcription regulatory circuitry driving microglia differentiation. (**A**) Master transcription factors predicted from the reconstruction of gene regulatory networks (GRN) and conserved in at least three over six of the studied protocols. Heatmap corresponds to their master regulatory index (MRI), which corresponds to the fraction of downstream target players (direct or indirect) that can be controlled by the TF within the reconstructed GRN (in percent). Transcription factors were organized into three groups: (a) those that were retrieved in the datasets issued from all protocols; (b) those retrieved from datasets generated by Chen et al. and at least two other protocols; and (c) those retrieved from at least three protocols except Chen’s protocol. The transcription factors SPI1 and CEBPA, which were over-expressed in the protocol described by Chen et al., are part of group (a), as highlighted by the “*”. (**B**) Cell ontology analysis was performed for each of the TF groups described in (**A**). (**C**) SPI/CEBPA co-regulatory network displaying their first TF neighbors within the master players listed in (**A**). The color-code associated with each node corresponds to the groups described in (**A**). Similarly, edges are colored in agreement with the color of the source node. (**D**) RNA-sequencing read count enrichment at the CEBPA locus for iMGL datasets issued from the different protocols (green) as well as the fetal and adult control samples (red). Notice the shorten read count enrichment pattern observed for the sample associated with the protocol described by Chen et al. relative to all other datasets. Enrichment read count levels were set to 300 for their enhanced comparability.

## Data Availability

Datasets reanalyzed in this study were collected from the Gene expression Omnibus (GEO) database. The corresponding GEO series accession numbers are: GSE97744, GSE85839, GSE97744, GSE117829, GSE169065 and GSE163984.

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
