# Peer review of "A Core Transcription Regulatory Circuitry Defining Microglia Cell Identity Inferred from the Reanalysis of Multiple Human Microglia Differentiation Protocols"

_brainsci, 2021, doi:10.3390/brainsci11101338_

Round 1

Reviewer 1 Report

In the manuscript “A core transcription regulatory circuitry defining microglia cell identity inferred from the reanalysis of multiple microglia differentiation protocols” the authors obtained transcriptomic dataset of iPSC derived microglia from different groups namely Abud et Al (2017) [26] (GSE97744), Muffat et al 67 (2016) [27] (GSE85839), Douvaras et al (2017) [22] (GSE97744), McQuade et al (2018) [21] 68 (GSE117829), Trudler et al (2021) [24] (GSE169065), and Chen et al (2021) [25] (GSE163984). The study is interesting but one major limitation is the availability of only a few datasets for comparisons of iPSC-derived microglia. However, more extensive analysis can be achieved with the publicly available dataset for brain resident microglia and monocyte-derived microglia.

My comments

  1. “When the iPSC data were not provided (Muffat et al (2016), Douvaras et al (2017) and McQuade et al (2018) data sets, iPSC datasets provided by Abud et al (2017) were used as a reference for the differential expression analysis”, this suggests that only 3 datasets were used for all the analysis.
  2. All the datasets are from the already published manuscripts. Why they used QC STAMP for quality control and what is the significance of that in this manuscript? What does low QC STAMP for Douvaras et al dataset means?
  3. Authors should include other datasets in their analysis Gosselin D et al, 2017, Grubman A et al, 2020, Banerjee A et al 2021. This will help to converge the data better as iPSC due to reprogramming may have some distinct profiles.  iPSC-derived microglia are modulated by the reprogramming cocktail, thus it is important to compare with datasets where no reprogramming is done. 
  4. In Fig 3 PCA plot, include dataset from brain resident and monocyte-derived microglia.

Author Response

Reviewer #1

Comments and Suggestions for Authors

In the manuscript “A core transcription regulatory circuitry defining microglia cell identity inferred from the reanalysis of multiple microglia differentiation protocols” the authors obtained transcriptomic dataset of iPSC derived microglia from different groups namely Abud et Al (2017) [26] (GSE97744), Muffat et al 67 (2016) [27] (GSE85839), Douvaras et al (2017) [22] (GSE97744), McQuade et al (2018) [21] 68 (GSE117829), Trudler et al (2021) [24] (GSE169065), and Chen et al (2021) [25] (GSE163984). The study is interesting but one major limitation is the availability of only a few datasets for comparisons of iPSC-derived microglia. However, more extensive analysis can be achieved with the publicly available dataset for brain resident microglia and monocyte-derived microglia.

We are glad to hear that reviewer #1 found our manuscript of major interest to the scientific community. This being said, we do not agree with the statement about the “few number” of datasets used in this study. Indeed, it is important to mention that our comparative effort has been performed over a total of 148 transcriptome datasets issued from six different protocols for microglia differentiation. While we have found further other datasets during the preparation of this manuscript, several of them were issued on mouse models or were generated with microarray technology, thus making it impossible to integrate them as this comparative study. For this reason, we have also changed the title of our manuscript, by including the term “human”, to precise that the TF regulatory circuit presented herein has been assessed only on human data.

Furthermore, as stated in the further comments, other types of datasets like those generated by Grubman et al, correspond per se to different terminal states; thus, a direct comparison between all six protocols studied on this manuscript with others might not lead to a consensus transcription regulatory circuitry.

My comments

“When the iPSC data were not provided (Muffat et al (2016), Douvaras et al (2017) and McQuade et al (2018) data sets, iPSC datasets provided by Abud et al (2017) were used as a reference for the differential expression analysis”, this suggests that only 3 datasets were used for all the analysis.

Considering that during this study we have focused on differential expression levels for performing comparisons, it was essential to count with iPSC datasets as reference. As properly stated, in cases in which the iPSC transcriptomes were not provided, we have used the four iPSC datasets provided by Abud et al as reference. As indicated in the provided supplementary table 1, there are 4 iPSC datasets provided by Abud et al; and two others provided by Trudler et al. It is important to stress that is surprising that the authors of these studies, published in major journals, failed to provide the corresponding iPSC control samples. This being said, the PCA analysis presented in Figure 3A clearly demonstrates that all 6 iPSC datasets, issued from two different studies, present strong similarities, thus excluding potential divergences between their backgrounds. 

All the datasets are from the already published manuscripts. Why they used QC STAMP for quality control and what is the significance of that in this manuscript? What does low QC STAMP for Douvaras et al dataset means?

While all datasets are issued from published articles, this does not exclude potential issues with the quality of the provided datasets. To illustrate this aspect, reviewer #1 can have a closer look at our database (NGS-QC: http://ngs-qc.org/database.php) in which is possible to retrieve public datasets having low-quality scores, which might directly impact the interpretation of the assessed readouts, notably due, for instance, to background issues, but also the low quality of reagents used during sample preparation, low sequencing depth in use, etc. (you can have a detailed illustration of such cases in our previous articles: Mendoza-Parra et al, NAR 2013; Mendoza-Parra et al, Genomics Data 2014; Mendoza-Parra et al, F1000 Res2016;  Blum et al,  Life Science Aliance 2019).

Furthermore, reviewer #1 might be aware of the initiatives launched by the scientific community concerning the low level of reproducibility in science (e.g. Is there a reproducibility crisis in science? Nature; May 2016), which becomes even harder to trace in the case of results issued from multiple computational processing over NGS data. Indeed, the quality of datasets in use should be assessed and provided together with raw data at the time of publication; an aspect that is not systematically performed by authors; reason why our team invested major efforts since 2013 for qualifying public datasets.

As stated in our manuscript, the first thing to do prior to performing dataset comparison si to verify their quality. Figure 1B confirms that all public datasets used in this study do present high-quality scores, as highlighted by the quality “A” threshold defined on the basis of our certification of more than 90 thousand public datasets (NGS-QC: http://ngs-qc.org/database.php).

Authors should include other datasets in their analysis Gosselin D et al, 2017, Grubman A et al, 2020, Banerjee A et al 2021. This will help to converge the data better as iPSC due to reprogramming may have some distinct profiles.  iPSC-derived microglia are modulated by the reprogramming cocktail, thus it is important to compare with datasets where no reprogramming is done.

We acknowledge reviewer #1 by suggesting to consider those studies. To address this point, we have looked carefully at each of the aforementioned articles. Indeed, Grubman et al (Stem cell Reports 2020) performed a similar effort like ours, by comparing public data concerning iPSC differentiation into microglia, to establish the baseline protocol leading to human microglia. Furthermore, they have studied strategies to enhance microglia maturation, notably by co-culturing with immortalized human neuronal precursors (ReN). As consequence, we consider that comparing all six protocols focused on generating microglia (or microglia precursors) with a matured protocol issued from a further co-culture step is not optimal; because per definition their terminal states (i.e. microglial precursors or matured microglia) are not the same.

This being said, we have included as part of the revised version of this manuscript, a new supplementary figure illustrating the fraction of master transcription factors shared between the consensus regulatory circuitry described in Figure 4 and those retrieved on the data provided by Grubman et al. (Supp. Figure S4). As part of this figure, we also include a comparison with the transcription factors revealed by Banerjee et al (Frontiers in cellular neuroscience; 2021), which correspond to a microglia-like cellular model derived from peripheral blood monocytes. Furthermore, we also included a comparison with the data provided by Gosselin et al; notably issued from human microglia isolated from surgically resected brain tissues, followed by their culture in in-vitro. Supp. Figure S4, clearly demonstrates that our consensus Master regulatory circuitry, described in figure4, shares TFs observed on these three studies, notably with those being in common with all six protocols, or those retrieved on at least three of them except on Chen’s protocol.

In Fig 3 PCA plot, include dataset from brain resident and monocyte-derived microglia.

In figure 3A we have included samples concerning fetal and adult microglia cells (labeled as “Microglia”), but also data corresponding to CD14+/CD16- monocytes (labeled as “CD14”), CD14+/CD16+ inflammatory monocytes (labeled as “CD16”), myeloid dendritic cells (“Blood DC”), neuronal precursors (“NPC”) and neuronal transcriptome data.

Reviewer 2 Report

The role of microglial activation in the neurodegenerative diseases is not fully explored. Moreover the access to factor enabling its assessment is limited. Authors of this study intended to analyze microglia differentiation protocols. In my opinion the work presents an interesting point of view however the study could be further improved:

1. Authors aknowledge the context of microglial activation in neurodegeneration in AD. It would be valuable to mention other examples as the 1 - latest review concerning another tauopathy PSP - Front Neurosc 2020 and 2 - synucleinopathies - Mechanisms of Neurodegeneration in Various Forms of Parkinsonism-Similarities and Differences - Koziorowski et al, 2021. The statement concerning microglial activation in neurodegenerative disorders should aknowledge the association between microglial activation and human endogenous retroviruses (HERV).

2. The discussion could benefit by adding a summarizing a table.

3. Limitations concerning this study should be clearly stated in a separate paragraph.

4. In the conclusion section authors could elaborate more widely on future perspectives.

Author Response

Reviewer #2:

Comments and Suggestions for Authors

The role of microglial activation in the neurodegenerative diseases is not fully explored. Moreover the access to factor enabling its assessment is limited. Authors of this study intended to analyze microglia differentiation protocols. In my opinion the work presents an interesting point of view however the study could be further improved:

We are glad to hear that reviewer #2 finds our manuscript of interest to the scientific community. We do agree with the fact that our manuscript does not directly address the role of microglia on neurodegenerative diseases, but this is basically due to the fact that microglial differentiation assays are in infancy. We expect that in the following years, scientists might provide transcriptome data concerning microglia differentiation from iPSC lines harboring neurodegenerative disease-related mutations. Hence, our current effort in defining a consensus master regulatory circuitry driving microglia differentiation will have a direct impact on the field, notably by its comparison with the data obtained from a neurodegenerative disease context.

  1. Authors aknowledge the context of microglial activation in neurodegeneration in AD. It would be valuable to mention other examples as the 1 - latest review concerning another tauopathy PSP - Front Neurosc 2020 and 2 - synucleinopathies - Mechanisms of Neurodegeneration in Various Forms of Parkinsonism-Similarities and Differences - Koziorowski et al, 2021. The statement concerning microglial activation in neurodegenerative disorders should aknowledge the association between microglial activation and human endogenous retroviruses (HERV).

Following the comments of reviewer #2, we have enhanced our introduction and discussion by taking into consideration the aforementioned references.

  1. The discussion could benefit by adding a summarizing a table.

We consider that the major message of our study is summarized in Figure 4, where we provide a view of the consensus transcription factors that are master regulators leading to microglia state.

  1. Limitations concerning this study should be clearly stated in a separate paragraph.

As stated in the comments discussed with reviewer #1; the focus of the study is to compare differentiation protocols leading to microglial state; thus, steps concerning microglia maturation, for instance, due to their co-culture (in a final phase) with immortalized neuronal precursors, are not detailed on this effort.

  1. In the conclusion section authors could elaborate more widely on future perspectives.

Following this comment, we have enhanced the revised version of our manuscript.

Round 2

Reviewer 1 Report

The authors tried to address the concerns and have included new data to further strengthen their manuscript. 

Reviewer 2 Report

I have no further comments.